# Actuator and Sensor Fault Reconstruction for Dynamic Positioning Vessels Based on Adaptive Unknown Input Observer

1st Jialiang Li
*College of Marine Electrical Engineering*
*Dalian Maritime University*
Dalian 116026, China
lijialiang@dlmu.edu.cn

2nd Yulong Tuo*
*College of Marine Electrical Engineering*
*Dalian Maritime University*
Dalian 116026, China
tuoyulong@dlmu.edu.cn

3rd Lingling Yu
*China Ship Development and Design Center*
Wuhan 430000, China
yulingling9230@163.com

4th Jingxiang Liu
*College of Marine Electrical Engineering*
*Dalian Maritime University*
Dalian 116026, China
jxliu@dlmu.edu.cn

5th Shasha Wang
*College of Marine Electrical Engineering*
*Dalian Maritime University*
Dalian 116026, China
wangshashadmu@dlmu.edu.cn

6th Zhouhua Peng
*College of Marine Electrical Engineering*
*Dalian Maritime University*
Dalian 116026, China
zhpeng@dlmu.edu.cn

*Abstract*—This paper proposes a actuator and sensor fault reconstruction (FR) strategy based on an adaptive unknown input observer (AUIO) for DP vessels affected by unknown external environmental disturbances. First, a mathematical model of the DP vessel with actuator and sensor faults is established. Then, an AUIO is designed to estimate the system state and reconstruct the fault information. Finally, simulation results show that the proposed AUIO can accurately estimate the system state and reconstruct actuator and sensor faults, even in the presence of unknown environmental disturbances.

*Index Terms*—dynamic positioning vessel, fault reconstruction, adaptive unknown input observer, actuator fault, sensor fault

## I. INTRODUCTION

Dynamic positioning (DP) vessels are widely used in ocean engineering due to their high degree automation and positioning precision. However, it is unavoidable that DP vessel experience faults in harsh sea conditions. If these faults are not promptly detected and addressed, they could lead to equipment damage, economic losses, environmental pollution, and even causing casualties. Therefore, to ensure timely identification of faults and mitigate their impact on performance, it is crucial to research and develop fault detection and diagnosis (FDD) [1]–[3] methods for DP vessels.

This work was supported in part by the National Key Research and Development Program of China (2022ZD0119902), in part by the National Natural Science Foundation of China (52101298, 52201409, 62273068, 51979020, 52271302, 52471374), in part by the Dalian Innovative Support Scheme for High-level Talents (2023RQ066), in part by the Fundamental Research Funds for the Central Universities (3132023508), in part by the Key Basic Research of Dalian (2023JJ11CG008), and in part by the Natural Science Foundation of Liaoning Province (2023-MS-120).

In FDD methods, the observer-based fault detection (FD) technique is widely applied. The core principle involves estimating the system state through an observer and then comparing these estimations with the actual measured values. If a significant difference arises between the state estimations by the observer and the actual measurements, it may indicate that a fault has occurred in the system. [3] proposes a computationally efficient observer-based distributed FD scheme for second-order networked control systems. [4] proposes a distributed FD method based on minimal-order observers for linear discrete-time stochastic multi-agent systems with Gaussian noises and multiple concurrent faults.

However, FD methods primarily focuses on detecting the occurrence and location of faults, and does not accurately estimate the magnitude and dynamic behavior of faults, hence it may not be suitable for complex systems that require precise control and compensation. Fortunately, observer-based fault reconstruction (FR) methods can provide the high-precision estimation information that the aforementioned mentioned FD methods cannot offer. [5] introduces a robust FR scheme utilizing a sliding mode observer (SMO) with nonlinear nominal models. [6] proposes an event-triggered FR approach using SMO for linear time-invariant systems. While the aforementioned FR methods based on SMO effectively estimate faults, they inherently introduce chattering, which compromises the accuracy of the estimation. [7] presents a robust and adaptive observer-based approach for reconstructing actuator faults in uncertain nonlinear systems. [8] proposes an iterative learning observer-based FR scheme for DP vessels with actuator faults. But the aforementioned FR methods based on observers only

consider actuator faults and do not consider sensor faults, implying that they are unable to estimate sensor faults. In addition, the design and functionality of the observer are based on the accurate acquisition of the actual system state information by the sensors. If a sensor fails, the estimation results provided by the observer will no longer be reliable. Therefore, it is necessary to design observer-based FR methods that consider both actuator faults and sensor faults.

Based on the above analysis, this paper proposes a novel fault reconstruction (FR) strategy for dynamic positioning (DP) vessels using an adaptive unknown input observer (AUIO) to address the challenges posed by unknown environmental disturbances, actuator faults, and sensor faults. This approach offers a significant advantage over traditional methods by accurately reconstructing both constant and time-varying faults. The incorporation of an $H_\infty$ performance index into the AUIO design further enhances the robustness of the FR process and mitigates the impact of unknown environmental disturbances. The observer gains are determined by sloving a linear matrix inequality optimization problem, thereby simplifying the design process. Finally, numerical simulations validate the effectiveness and feasibility of the proposed method, demonstrating its potential to improve the safety and reliability of DP vessels. The main contributions of this paper are as follows:

- Compared to the FR method in [8], [9], this paper incorporates the $H_\infty$ performance index into the design of AUIO, thereby ensuring that the FR method based on AUIO has strong robustness. Even in the presence of external environmental disturbances, this method can still accurately reconstruct the actuator and sensor faults.
- In contrast to the FR method in [10]–[12], which only accounts for actuator faults, the proposed FR method based on AUIO considers both actuator and sensor faults. This enables the effective simultaneous reconstruction of both types of faults.

## II. MATHEMATICAL MODEL OF DP VESSEL WITH ACTUATOR AND SENSOR FAULT

To depict the motion of the DP vessel, Fig.1 presents the North-East-down (NED) and body-fixed coordinate frames. In this illustration, $O_N - X_N$, $O_N - Y_N$ and $O_N - Z_N$ represent north, east, and down directions, respectively; while $O_B - X_B$, $O_B - Y_B$ and $O_B - Z_B$ denote the fore, starboard, and bottom directions of the hull.

Considering the DP vessel control system model with unknown external environmental disturbances, the DP vessel kinematics and dynamics models are established as follows ([13]):

$$\begin{cases} \dot{\eta} = R(\psi)v \\ M\dot{v} + C(v)v + D(v)v = \tau + \tau_{env} \end{cases}, \quad (1)$$

where $\eta = [x, y, \psi]^T \in R^3$ represents the position and heading vector in the NED frame, while $\nu = [u, v, r]^T \in R^3$ denotes the velocity vector in the body-fixed frame, $M$ represents the

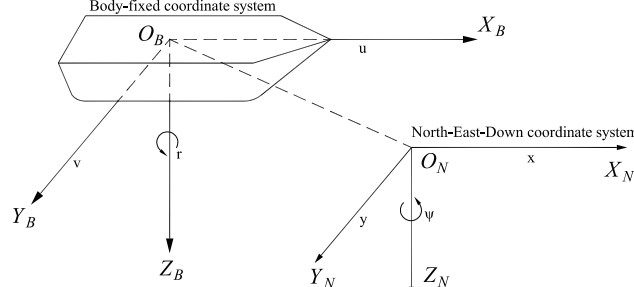

Figure 1. North-East-down (NED) and body-fixed coordinate frames of a vessel.

inertia matrix, $C(\nu) \in R^{3 \times 3}$ expresses the Coriolis-centripetal matrix, and $D(\nu) \in R^{3 \times 3}$ denotes the damping coefficient matrix. More details are provided in [13]. $\tau \in R^3$ represents the force and torque vector exerted on the DP vessel in the body-fixed frame, $\tau_{env} \in R^3$ is the unknow environmental disturbances acting on DP vessel, $R(\psi) \in R^{3 \times 3}$ is the rotation matrix and $R^{-1} = R^T$.

**Assumption 1.** *The desired trajectory $\eta_d = [x_d, y_d, \psi_d]^T$ is differentiable with bounded $\eta, \dot{\eta}$ and $\ddot{\eta}$.*

**Assumption 2.** *The external environmental disturbance $\tau_{env}$ is bounded.*

To facilitate the subsequent design of the AUIO and make the expression clearer, (1) is rewritten as following:

$$\ddot{\eta} = f(\eta, \dot{\eta}) + RM^{-1}\tau + RM^{-1}\tau_{env}, \quad (2)$$

where $f(\eta, \dot{\eta}) = -RM^{-1}[C(\nu) + D(\nu) - MR^T\dot{R}]R^T\dot{\eta}$.

In DP vessels, when actuators fail, there is a discrepancy $\Delta\tau$ between the actual torque $\tau$ applied to the thrusters and the commanded input torque $\tau_c$. This discrepancy can be expressed by the following formula:

$$\tau = \tau_c + \Delta\tau, \quad (3)$$

where $\Delta\tau$ is the thruster fault deviation torque.

Taking into account the actuator faults and sensor faults in DP vessels, and considering the actual outputs of position $\eta$ and its derivatives $\dot{\eta}$ as measured by the sensors, (2) is revised as follows:

$$\begin{cases} \dot{X} = aX + g(t, X) + bRM^{-1}\tau_c + F_a f_a(t) + cRM^{-1}\tau_{env} \\ Y = X + F_s f_s(t) \end{cases}, \quad (4)$$

where $X = [\eta^T \ \dot{\eta}^T]^T$ represents system state vector, $Y$ is the actual system output measured by the sensors; $g(t, X) = [0_{1 \times 3} \ f(\eta, \dot{\eta})^T]^T$ represents the nonlinear term in system (4) that satisfies the Lipschitz condition, i.e., satisfies the inequality $\| g(t, \hat{X}) - g(t, X) \| \le \| L_g(\eta(t) - \hat{\eta}(t)) \|$, where $\hat{X}$ is the estimation reconstructed by AUIO, $L_g \in R^{6 \times 6}$ is Lipschitz constant matrix and $g(0, 0) = 0$; $f_a(t) = RM^{-1}\Delta\tau$ represents the actuator faults and $f_s(t)$ represents the sensor faults. $a = \begin{bmatrix} 0_{3 \times 3} & I_{3 \times 3} \\ 0_{3 \times 3} & 0_{3 \times 3} \end{bmatrix}$ and $b = c = F_a = F_s = \begin{bmatrix} 0_{3 \times 3} \\ I_{3 \times 3} \end{bmatrix}$ are constant matrices.

## III. DESIGN OF AUIO

In order to reconstruct sensor faults, the system (4) will be next rewritten in the form of a singular system. Firstly, let $e = [I_{6\times6} \ 0_{6\times3}]$, $m = [a \ 0_{6\times3}]$ and $h = [I_{6\times6} \ F_s]$.

Since $\begin{bmatrix} e \\ h \end{bmatrix}$ is a column full-rank matrix,

matrix $\left( \begin{bmatrix} e \\ h \end{bmatrix}^T \begin{bmatrix} e \\ h \end{bmatrix} \right)^{-1}$ exists. Let $[p \ q] = \left( \begin{bmatrix} e \\ h \end{bmatrix}^T \begin{bmatrix} e \\ h \end{bmatrix} \right)^{-1} \begin{bmatrix} e \\ h \end{bmatrix}^T$, then it follows that:

$$pe + qh = I. \tag{5}$$

Define the vector $\xi = [X^T \ f_s^T]^T$, and the system (4) can be rewritten as follows:

$$\begin{cases} e\dot{\xi} = m\xi + g(t, e\xi) + bRM^{-1}\tau_c + F_a f_a(t) + cRM^{-1}\tau_{env} \\ Y = h\xi \end{cases} \tag{6}$$

In order to reconstruct the actuator faults $f_a$ and sensor faults $f_s$ in system (6), the following AUIO is designed:

$$\begin{cases} \dot{z} = nz + lY + p\hat{g}(t) + pbRM^{-1}\tau_c + pF_a\hat{f}_a \\ \hat{\xi} = z + qY \\ \hat{Y} = h\hat{\xi} \\ \dot{\hat{f}}_a = -s\tilde{Y} \end{cases} , \tag{7}$$

where $z$ is the observer (7) state, $\hat{Y}$ is the estimation of system (6) output $Y$, $\tilde{Y} = \hat{Y} - Y$ is the output estimation error, $\hat{g}(t) = g(t, e\hat{\xi})$ denotes the estimation of $g(t, e\xi)$, $\hat{\xi}$ and $\hat{f}_a$ represent the state estimation and actuator faults estimation of system (6), respectively. $n, l$ and $s$ are the observer gain matrices to be designed later. From formula (7), we can obtain the position estimation $\hat{\eta}$ and velocity estimation $\hat{\nu}$ for the DP vessel reconstructed by AUIO as follows: $\hat{\eta} = [\hat{\xi}_1 \ \hat{\xi}_2 \ \hat{\xi}_3]^T$, $\hat{\nu} = R^T(\hat{\eta})[\hat{\xi}_4 \ \hat{\xi}_5 \ \hat{\xi}_6]^T$, and $\hat{\xi}_i$ represents the ith element of vector $\hat{\xi}$.

Let $\tilde{\xi} = \hat{\xi} - \xi$ and $\tilde{f}_a = \hat{f}_a - f_a$, it can be known from formulas (5)~(7) that $\tilde{\xi} = \hat{\xi} - \xi = z + qh\xi - \xi = z - pe\xi$. And the derivative of $\tilde{f}_a$ is:

$$\dot{\tilde{f}}_a = \dot{\hat{f}}_a - \dot{f}_a = -s\tilde{Y} - \dot{f}_a = -sh\tilde{\xi} - \dot{f}_a. \tag{8}$$

Then, differentiate with respect to $\tilde{\xi}$ to obtain the following formula:

$$\begin{aligned} \dot{\tilde{\xi}} &= \dot{z} - pe\dot{\xi} \\ &= nz + lY + p\hat{g}(t) + pbRM^{-1}\tau_c + pF_a\hat{f}_a - pe\dot{\xi} \\ &= nz + lY + p\hat{g}(t) + pbRM^{-1}\tau_c + pF_a\hat{f}_a \\ &\quad - p[m\xi + g(t) + bRM^{-1}\tau_c + F_a f_a + cRM^{-1}\tau_{env}] \\ &= nz + lY + p[\tilde{g}(t) + F_a\tilde{f}_a - m\xi - cRM^{-1}\tau_{env}] \\ &= n[\tilde{\xi} + \xi - qY] + lY - p[cRM^{-1}\tau_{env} + \tilde{g}(t) + F_a\tilde{f}_a \\ &\quad - m\xi] \\ &= n\tilde{\xi} + (n - pm)\xi + (l - nq)Y + p[\tilde{g}(t) + F_a\tilde{f}_a \\ &\quad - cRM^{-1}\tau_{env}] \\ &= n\tilde{\xi} + (n - pm)\xi + (l - nq)h\xi + p[\tilde{g}(t) + F_a\tilde{f}_a \\ &\quad - cRM^{-1}\tau_{env}], \end{aligned} \tag{9}$$

where $\tilde{g}(t) = \hat{g}(t) - g(t)$.

To make the derivation more concise, the following formulas are assumed:

$$\begin{cases} F = l - nq \\ N = pm - Fh \end{cases} . \tag{10}$$

Substituting (10) into (9), the following formula cam be obtained:

$$\begin{aligned} \dot{\tilde{\xi}} &= n\tilde{\xi} + (n - pm)\xi + Fh\xi + p[\tilde{g}(t) + F_a\tilde{f}_a - cRM^{-1}\tau_{env}] \\ &= n\tilde{\xi} + (n - pm + Fh)\xi + p[\tilde{g}(t) + F_a\tilde{f}_a - cRM^{-1}\tau_{env}] \\ &= (pm - Fh)\tilde{\xi} + p[\tilde{g}(t) + F_a\tilde{f}_a - cRM^{-1}\tau_{env}] \\ &= N\tilde{\xi} + p[\tilde{g}(t) + F_a\tilde{f}_a - cRM^{-1}\tau_{env}]. \end{aligned} \tag{11}$$

Define $\varepsilon = [\tilde{\xi} \ \tilde{f}_a]^T$ and $d = [(RM^{-1}\tau_{env})^T \ \dot{f}_a^T]^T$, and substituting (8) into (11), the following conclusion can be obtained:

$$\dot{\varepsilon} = A\varepsilon + B + Ed, \tag{12}$$

where $A = \begin{bmatrix} N & pF_a \\ -sh & 0 \end{bmatrix}$, $B = \begin{bmatrix} p\tilde{g}(t) \\ 0 \end{bmatrix}$ and $E = \begin{bmatrix} -pc & 0 \\ 0 & -I \end{bmatrix}$.

Let $A_1 = \begin{bmatrix} pm & pF_a \\ 0 & 0 \end{bmatrix}$, $A_2 = \begin{bmatrix} h & 0 \end{bmatrix}$ and $Q = [F^T s^T]^T$, then $A = A_1 - QA_2$.

To suppress the impact of unknown environmental disturbances $\tau_{env}$ on FR based on AUIO, design an $H_\infty$ performance index $\gamma > 0$ such that:

$$\| \varepsilon \| \le \gamma \| d \| . \tag{13}$$

Since $\| \varepsilon \| \ge \| \tilde{f}_a \|$ and $\| \varepsilon \| \ge \| \tilde{f}_s \|$, the proposed AUIO can achieve robust asymptotic reconstruction of actuator and sensor faults when formula (13) holds. It can be seen from formula (13) that a smaller $\gamma$ will result in a smaller error in FR. By solving the optimization problem in **Theorem 1**, the minimum of $\gamma$ can be obtained, ensuring that the error dynamic system (12) is robustly asymptotically stable.

**Theorem 1.** *Considering system (6) and ADIO system (7), if there exist positive definite matrix $P_1$ and matrix $P_2$ such*

that the following matrix inequality optimization problem (14) holds, then the error dynamic System (12) is robustly asymptotically stable.

$$min \ \gamma,$$
$$s.t. \ \gamma > 0,$$

$$\Gamma = \begin{bmatrix} \bar{\Gamma} & P_1 & P_1E & I_{12\times12} \\ * & -I_{12\times12} & 0 & 0 \\ * & * & -\gamma I_{6\times6} & 0 \\ * & * & * & -\gamma I_{12\times12} \end{bmatrix} < 0, \quad (14)$$

where $\bar{\Gamma} = P_1 A_1 - P_2 A_2 + A_1^T P_1 - A_2^T P_2^T + A_g$, $A_g = \begin{bmatrix} \| p \| (L_g e)^T L_g e & 0 \\ 0 & 0 \end{bmatrix}$ and $*$ represents the symmetric term of the symmetric matrix.

*Proof:* Define the Lyapunov function $V = \varepsilon^T s \gamma \varepsilon$ and take its derivative to obtain:

$$
\begin{aligned}
\dot{V} &= \varepsilon^T s\gamma\dot{\varepsilon} + \dot{\varepsilon}^T s\gamma\varepsilon \\
&= \varepsilon^T s\gamma(A\varepsilon + B + Ed) + (A\varepsilon + B + Ed)^T s\gamma\varepsilon \\
&= \varepsilon^T(s\gamma A + A^T s\gamma)\varepsilon + \varepsilon^T s\gamma E + E^T s\gamma\varepsilon + \varepsilon^T s\gamma Ed \\
&\quad + d^T E^T s\gamma\varepsilon.
\end{aligned}
\quad (15)
$$

In addition, it can be calculated that:

$$
\begin{aligned}
\begin{bmatrix} p[\tilde{g}(t)] \\ 0 \end{bmatrix} &\leq \| p \| \| \tilde{g}(t) \| = \| p \| [\hat{g}(t) - g(t)]^T[\hat{g}(t) - g(t)] \\
&\leq \| p \| [\hat{\xi} - \xi]^T e^T L_g^T L_g e[\hat{\xi} - \xi] \\
&= \| p \| [L_g e[I_{9\times9} \ 0]\varepsilon]^T L_g e[I_{9\times9} \ 0]\varepsilon \\
&= \| p \| \varepsilon^T[L_g e \ 0]^T[L_g e \ 0]\varepsilon \\
&= \varepsilon^T \begin{bmatrix} \| p \| (L_g e)^T L_g e & 0 \\ 0 & 0 \end{bmatrix} \varepsilon \\
&= \varepsilon^T A_g \varepsilon.
\end{aligned}
\quad (16)
$$

Therefore, the following formula is obtained:

$$\varepsilon^T A_g \varepsilon - B^T I_{12\times12} B \geq 0. \quad (17)$$

Let $J = \int_0^\infty \frac{\varepsilon^T\varepsilon - \gamma^2 d^T d}{\gamma} dt$, thereby the following formula is got:

$$J < \int_0^\infty \frac{\varepsilon^T\varepsilon - \gamma^2 d^T d + \dot{V}}{\gamma} dt = \bar{J}. \quad (18)$$

From formulas (15), (17), and (18), it can be deduced that:

$$
\begin{aligned}
\bar{J} &\leq \frac{1}{\gamma}[\varepsilon^T\varepsilon - \gamma^2 d^T d + \varepsilon^T(P_1\gamma A + A^T P_1\gamma)\varepsilon \\
&\quad + \varepsilon^T P_1\gamma B + B^T P_1\gamma\varepsilon + \varepsilon^T P_1\gamma Ed + d^T E^T P_1\gamma\varepsilon] \\
&\quad + \varepsilon^T A_g\varepsilon - B^T I_{12\times12} B \\
&\leq \varepsilon^T\left(P_1 A + A^T P_1 + A_g + \frac{1}{\gamma}I_{12\times12}\right)\varepsilon \\
&\quad + \varepsilon^T PB + B^T P\varepsilon + \varepsilon^T P_1 Ed + d^T E^T P\varepsilon - \gamma d^T d \\
&\quad - B^T I_{12\times12} B.
\end{aligned}
\quad (19)
$$

Let $G = [\varepsilon^T \ B^T d^T]^T$, and it can be concluded that $\bar{J} = G^T \Omega G$ from formula (19), where

$$\Omega = \begin{bmatrix} P_1 A + A^T P_1 + A_g + \frac{1}{\gamma}I_{12\times12} & P_1 & P_1 E \\ * & -I_{12\times12} & 0 \\ * & * & -\gamma I_{12\times12} \end{bmatrix}. \quad (20)$$

And $\Omega < 0$ is equivalent to the following inequality can be got by using the Schur complement theorem:

$$\Gamma = \begin{bmatrix} P_1 A + A^T P_1 + A_g & P_1 & P_1 E & I_{12\times12} \\ * & -I_{12\times12} & 0 & 0 \\ * & * & -\gamma I_{6\times6} & 0 \\ * & * & * & -\gamma I_{12\times12} \end{bmatrix}. \quad (21)$$

Let $P_3 = P_1 Q$, then $PA = PA_1 - P_3 A_2$, therefore formula (21) can be rewritten as following :

$$\Gamma = \begin{bmatrix} \bar{\Gamma} & P_1 & P_1 E & I_{12\times12} \\ * & -I_{12\times12} & 0 & 0 \\ * & * & -\gamma I_{6\times6} & 0 \\ * & * & * & -\gamma I_{12\times12} \end{bmatrix} < 0, \quad (22)$$

If inequality (22) has a solution, then $\bar{J} < 0$, and consequently $J < 0$ and $\| \varepsilon \| \leq \gamma \| d \|$. ∎

## IV. SIMULATION RESULTS ANALYSIS

To verify the effectiveness of the proposed FR method based on the AUIO, this chapter utilizes the DP ship model as a verification case. To verify the effectiveness of the proposed FR method based on the AUIO, this section conducts simulation verification using the "Hai Yang Shi You 201" [13] DP vessel model as a test case.

The simulation is set to run for a total of 500 seconds. To maintain stable motion for the DP vessel, the control input $\tau_c$ of the DP system is designed as sliding mode control. The initial states of the DP vessel are set to zero. Additionally, the desired trajectory is specified as $\eta_d(t) = [t \ m, 100\sin(t/100) \ m, \pi \ rad]^T$. The unknown environmental disturbance is set as the marine environmental disturbance in [14], with the wind speed set to $8m/s$ and the current velocity set to $1.5m/s$. All thrusters and sensors of the DP vessel are in healthy when $0 \leqslant t < 50s$, and the thruster fault deviation torque $\Delta\tau$ and sensor fault $f_s(t)$ are set as follows when $t \geqslant 50s$:

$$\Delta\tau = \begin{cases} = 10^6 \times [2sin(t/50)cos(t/100) + 3] \\ = 2 \times 10^6 \\ = 10^6 \times [10sin(3t/100)cos(t/100) - 15] \end{cases},$$

$$f_s = \begin{cases} = 100 \times [5(1 - exp(10 - t/5)) + sin(\pi t/50)] \\ = 500 \\ = 100 \times [5(1 - exp(4 - t/15)) + cos(\pi t/250)] \end{cases}.$$

By solving the linear matrix inequality (14), we obtain $\gamma_{min} = 14.3207$. Subsequently, this allows us to derive the observer gain matrices $n, l$ and $s$ for observer (7).

Fig.2 ∼ Fig.5 show the state estimation curves and state estimation error curves for the DP vessel system (1) reconstructed by the proposed AUIO. It can be seen from Fig.2 and Fig.4 that the proposed AUIO can quickly track the state of the DP vessel. Fig.3 and Fig.5 demonstrate that the state estimation errors $\tilde{\eta}$ and $\tilde{\nu}$ remain within a narrow range. Additionally, even when faults occur at $t = 50s$, the estimation errors for both position and velocity converge quickly.

Fig.6 ∼ Fig.9 display the FR estimation curves and the corresponding estimation error curves for the actuator faults and sensor faults. Fig.6 ∼ Fig.9 illustrate that the proposed AUIO-based FR method maintains precision and robustness in simultaneously estimating actuator faults and sensor faults within the DP vessel system (1), even in the presence of unknown environmental disturbances $\tau_{env}$. Fig.6 illustrates the reconstruction capability of the AUIO for time-varying actuator faults $f_{a1}$ and $f_{a3}$, as well as the slow-varying actuator fault $f_{a2}$. Furthermore, it is evident from Fig.6 and Fig.7 that for the abrupt actuator faults that occur at $t = 50s$, the proposed AUIO is able to quickly reconstruct the actuator faults with negligible impact. Fig.8 shows that the proposed AUIO has the capability to reconstruct periodic time-varying sensor faults $f_{s1}$, constant sensor faults $f_{s2}$, and slow-varying sensor faults $f_{s3}$. Fig.8 and Fig.9 also demonstrate that when abrupt sensor faults occur at $t = 50s$, the proposed AUIO can rapidly detect this fault and quickly track the fault signal.

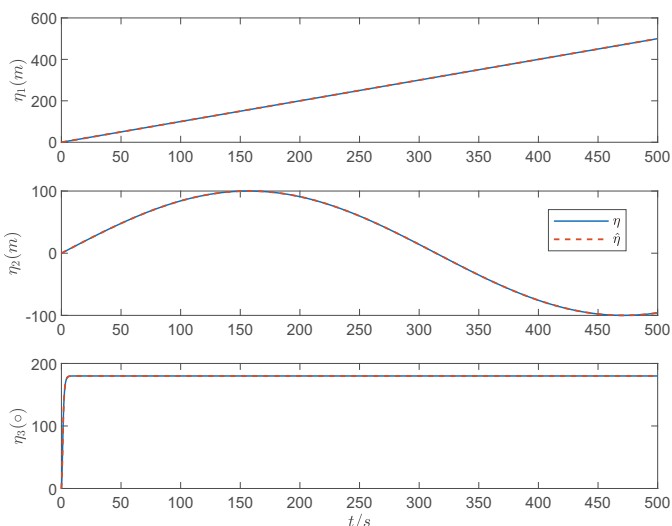

Figure 2.  The position $\eta$ of the DP vessel and its estimation $\hat{\eta}$

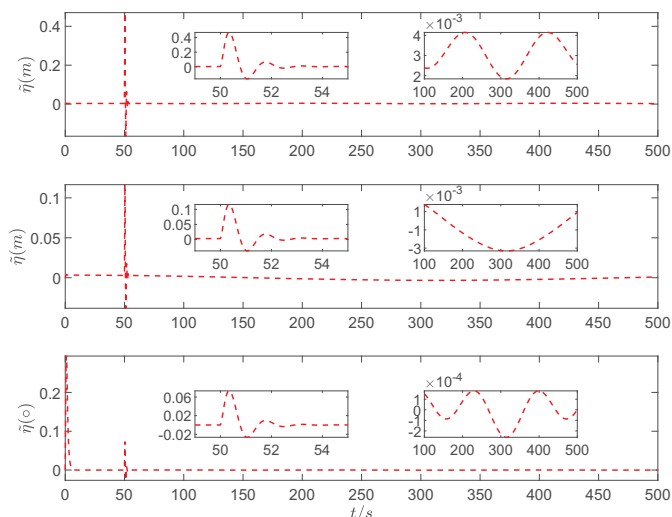

Figure 3.  The position estimation error $\tilde{\eta}$ of the DP vessel

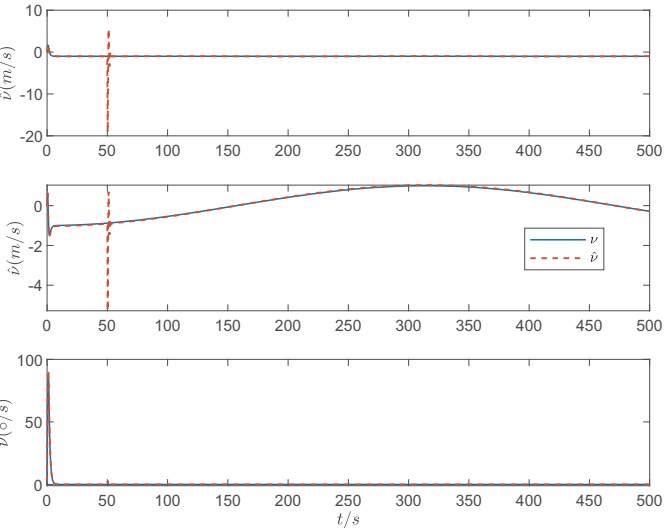

Figure 4.  The velocity $\nu$ of the DP vessel and its estimation $\hat{\nu}$

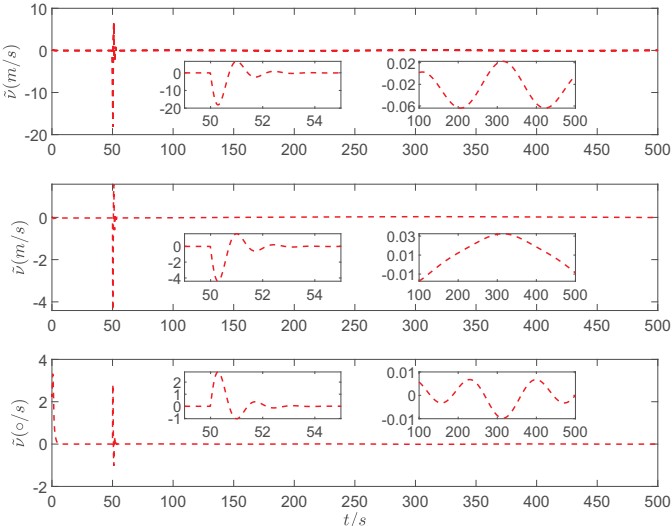

Figure 5.  The velocity estimation error $\tilde{\nu}$ of the DP vessel

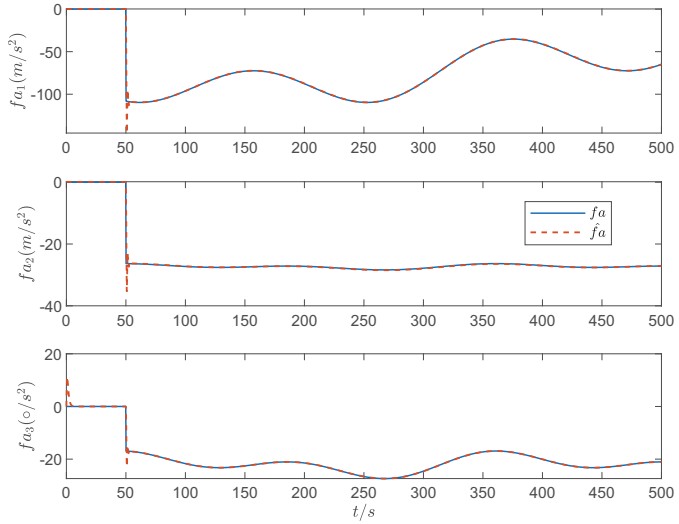

Figure 6.   The actuator fault $f_a$ and its estimation $\hat{f}_a$

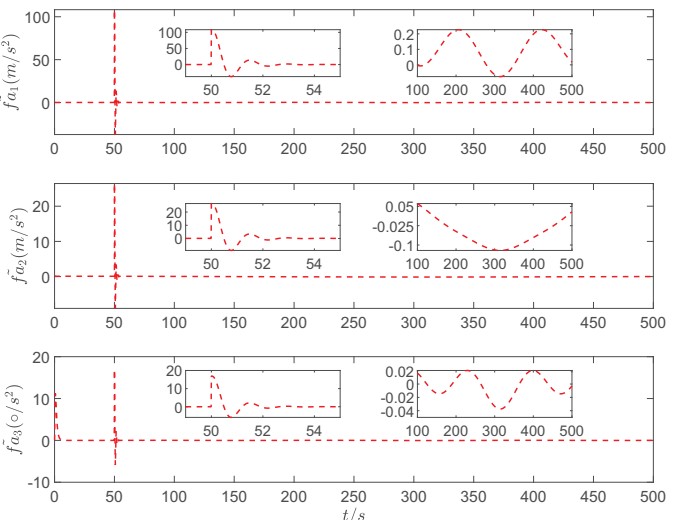

Figure 7.   The actuator fault estimation error $\tilde{f}_a$

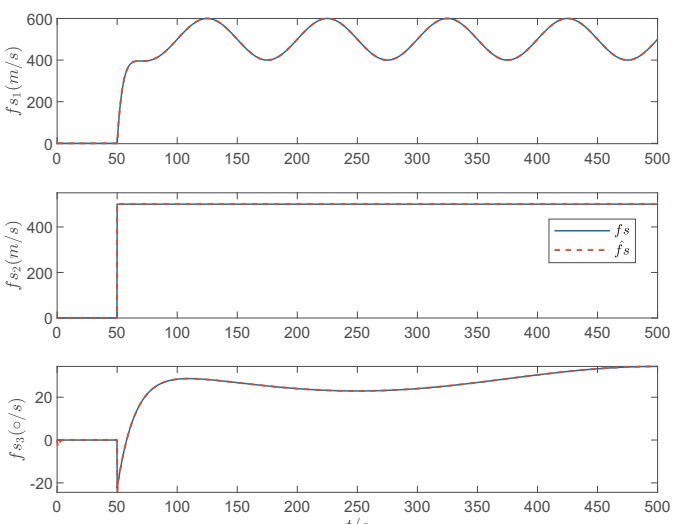

Figure 8.   The senor fault $f_s$ and its estimation $\hat{f}_s$

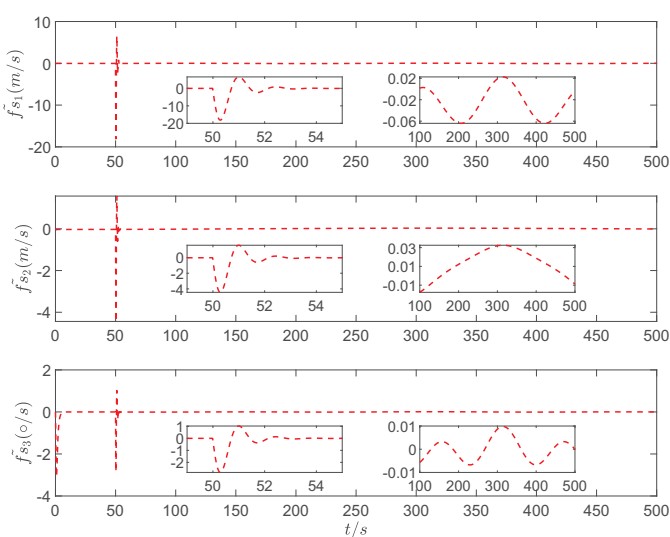

Figure 9.   The senor fault estimation error $\tilde{f}_s$

## V. CONCLUSIONS

This paper presents a novel FR strategy for DP vessels affected by unknown external environmental disturbances based on an AUIO, considering both actuator faults and sensor faults. The proposed method effectively estimates the system state and reconstructs fault information, even in the presence of unknown environmental disturbances. The simulation results validate the proposed method's effectiveness and robustness in estimating constant and various time-varying faults, indicating its potential for ensuring the safety and reliability of DP vessel in practical applications. Future work will focus on integrating this method into fault-tolerant control strategies and improving observer convergence performance.

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
