# OpenReview forum: "Actuator and Sensor Fault Reconstruction for Dynamic Positioning Vessels Based on Adaptive Unknown Input Observer"
_IEEE.org/ICIST/2024/Conference — IEEE ICIST 2024 Conference Submission_

### Official Review · Reviewer_kzNT · 2024-08-24
**The paper is written clearly, exceptionally excellent.**

**Rating:** 8
**Confidence:** 3

**Review:**

This paper excels in terms of quality, clarity, originality, and significance, but I would still like to offer some suggestions.
1. In the paper, what assumptions were made, and explain their rationality.
2. What original contributions should the thesis emphasize?

---

### Official Review · Reviewer_e7T8 · 2024-08-25
**summarize the innovation  and individual originality work point by point**

**Rating:** 7
**Confidence:** 4

**Review:**

For the paper titled "Actuator and Sensor Fault Reconstruction for Dynamic Positioning Vessels Based on Adaptive Unknown Input Observer", it is suggested to summarize the innovation  and individual originality work point by point, in other words, add the content of this section.It is recommended to add sufficient  analysis and complete interpretation of all data to prove the rationality of the conclusion and add sufficient background and literature comparison.

---

### Official Review · Reviewer_MbfL · 2024-08-25
**The work content of this paper is to design an audio-based system state estimation and fault information reconstruction method for DP ships disturbed by unknown external environment. The method can accurately estimate the system state and reconstruct actuator and sensor faults, even in the face of unknown environmental interference. However, 1. It is necessary to emphasize the original innovation work. 2. What assumptions are used and the reasonableness of these assumptions needs to be explained.**

**Rating:** 7
**Confidence:** 4

**Review:**

The work content of this paper is to propose a fault reconstruction strategy for actuators and sensors based on adaptive unknown input observers (AUIOs) for DP ships disturbed by unknown external environments. A mathematical model of a DP ship with faulty steering gear and sensors was established. The solution used is to design an audio-based system state estimation and fault information reconstruction method. The method can accurately estimate the system state and reconstruct actuator and sensor faults, even in the face of unknown environmental interference.
However, 1. It is necessary to emphasize the original innovation work.
2. What assumptions are used and the reasonableness of these assumptions needs to be explained.

---

### Decision · Program_Chairs · 2024-09-08

Accept (Oral)